# Oncolytic Vaccinia Virus Harboring *Aphrocallistes vastus* Lectin Inhibits the Growth of Cervical Cancer Cells Hela S3

**DOI:** 10.3390/md19100532

**Published:** 2021-09-23

**Authors:** Jiajun Ni, Hualin Feng, Xiang Xu, Tingting Liu, Ting Ye, Kan Chen, Gongchu Li

**Affiliations:** College of Life Sciences and Medicine, Zhejiang Sci-Tch University, Hangzhou 310018, China; nijj@mails.zstu.edu.cn (J.N.); fenghl@mails.zstu.edu.cn (H.F.); xux@mails.zstu.edu.cn (X.X.); liutt@mails.zstu.edu.cn (T.L.); yeting@zstu.edu.cn (T.Y.)

**Keywords:** *Aphrocallistes vastus* lectin, sphere formation assay, oncolytic vaccinia virus, Hela S3 cell line

## Abstract

*Aphrocallistes vastus* lectin (AVL) is a C-type marine lectin produced by sponges. Our previous study demonstrated that genes encoding AVL enhanced the cytotoxic effect of oncolytic vaccinia virus (oncoVV) in a variety of cancer cells. In this study, the inhibitory effect of oncoVV-AVL on Hela S3 cervical cancer cells, a cell line with spheroidizing ability, was explored. The results showed that oncoVV-AVL could inhibit Hela S3 cells growth both in vivo and in vitro. Further investigation revealed that AVL increased the virus replication, promote the expression of OASL protein and stimulated the activation of Raf in Hela S3 cells. This study may provide insight into a novel way for the utilization of lection AVL.

## 1. Introduction

Vaccinia virus is a large, enveloped, double-stranded DNA virus with a linear genome approximately 190 kb in length. It has been widely used in the eradication of smallpox [1]. Oncolytic vaccinia virus harboring therapeutic genes has been applied to lyse tumor cells directly [2,3,4]. To date, several oncolytic vaccinia viruses derived from Wyeth, Western Reserve, and Lister vaccinia strains are being used clinically [5,6,7]. The strategies to improve the oncolytic efficacy of vaccinia vectors have become the focus of research.

In recent years, the role of the marine environment in the development of anticancer drugs has been widely reviewed. The compounds derived from marine organisms may be an inspiring tool to develop new anticancer agents [8]. Lectins are widely distributed in marine bioresources, such as marine cyanobacteria, algae, invertebrate animals and fish. Previous studies demonstrated that lectins had been developed by a variety of biological techniques by binding with carbohydrates, such as lectin array, lectin blot and lectin-based chromatography [9,10,11,12,13].

*Aphrocallistes vastus* lectin (AVL) is a Ca^2+^ dependent C-type lectin produced by sponges [14]. In our previous study, a gene encoding AVL was carried by an oncolytic vaccinia virus (oncoVV) vector, which is deficient of the TK gene for cancer specific replication, forming a recombinant virus oncoVV-AVL. We demonstrated that oncoVV-AVL enhanced the inhibitory effect of oncolytic vaccinia virus in a variety of cancer cells, such as colorectal cancer line HCT116, glioma cell line U251 and hepatocellular carcinoma cell line BEL-7404 [15]. However, the role of oncolytic vaccinia virus oncoVV-AVL in cervical cancer cells, especially the cells with stem characteristics, remains uncertain.

In this study, we investigated the inhibitory effect of oncoVV-AVL on cervical cancer cell line Hela S3, a tumor cell with spheroidizing characteristics, and further analyzed the underlying mechanism.

## 2. Results

### 2.1. Tumorospheres Formation Assay

To compare the oncogenic property between Hela and Hela S3 cells, a tumor sphere formation assay was performed. The results showed that Hela S3 cells continuously formed spheres through several subsequent serial passages, whereas Hela cells did not show the ability to form spheres (Figure 1a,b). Previous studies suggested that sphere- forming activity represented the stemness of the cells to a certain extent [16,17]. Hela S3 cells exhibited the property of stemness in vitro. Thus, Hela S3 cells were studied in the following study.

### 2.2. The Cytotoxic Effect of OncoVV-AVL on Cervical Cancer Cell Line Hela S3

A MTT assay was performed to examine the cytotoxic effect of oncoVV-AVL on Hela S3 and Hela cells. Cells were infected with oncoVV or oncoVV-AVL, and cell viability was measured after 24, 48, 72 and 96 h, respectively (Figure 2a,b). The results indicated that oncoVV-AVL elicited a significantly cytotoxic effect in dose and time dependent manners as compared with oncoVV. These suggested that AVL could enhance the cytotoxic effect of the oncolytic vaccinia virus.

To confirm the cytotoxic effect of oncoVV-AVL on Hela S3 cells, flow cytometry was performed to detect the apoptotic/dead cells infected by oncoVV or oncoVV-AVL. As shown in Figure 2c, compared with oncoVV and PBS controls, oncoVV-AVL induced a significantly higher level of cytotoxicity. However, there was no significant difference in the percentage of AnnexinV-FITC positive apoptotic cells among these groups, suggesting that AVL promoted nonapoptotic cell death induced by the oncolytic vaccinia viruses.

### 2.3. OncoVV-AVL Inhibited the Growth of Tumorospheres in Hela S3 Cells

To further investigate the cytotoxic effect of oncoVV-AVL on tumor cells with stemness, the spheres of Hela S3 cells were infected by oncoVV or oncoVV-AVL at the concentration of 5 MOI. Seven days later, the morphology of tumorosphere was observed under microscope. As shown in Figure 3a, the spheres in the oncoVV-AVL group became smaller, dark and scattered as compared with the oncoVV group. Meanwhile, the number of tumorospheres in the oncoVV-AVL group was dramatically lower than that in the oncoVV group (Figure 3b). This demonstrated that oncoVV-AVL inhibited the growth of tumorospheres of the Hela S3 cells.

### 2.4. AVL Harboring Improved the Replication of Oncolytic Vaccinia Viruses in Hela S3 Cells

We further explored the underlying mechanisms of oncoVV-AVL killing Hela S3 cells. Firstly, TCID50 method was used to test the replications of oncoVV-AVL and oncoVV in Hela S3 cells at the time points of 0, 12, 24, 36 and 48 h, respectively. The results showed that the reproductive number of oncoVV-AVL was significantly higher than that of oncoVV (Figure 4a).

Oligoadenylate synthase-like protein (OASL) is a member of OAS family proteins without OAS enzyme activity. Previous studies reported that OASLs from different mammalian exhibited antiviral activity [18,19,20]. Ghosh’s study determined that OASL elevated DNA virus replication, including herpes simplex virus (HSV), adenovirus and vaccinia virus [21]. In this study, the expression of OASL in Hele S3 cells was detected using Western blot. We found that OASL protein was obviously expressed in oncoVV or oncoVV-AVL infected cells, but not in PBS treated cells. Furthermore, the level of OASL in oncoVV-AVL group was much higher than that in oncoVV group. This suggested that AVL may enhance the replication of oncolytic vaccinia virus by elevating the expression of the OASL protein.

### 2.5. OncoVV-AVL Infection Altered the Raf/ERK Signaling Pathway in Hela S3 Cells

The mitogen-activated protein kinase (MAPK) signaling pathway is ubiquitous in a variety of biological cells and is activated in the majority of advanced tumors. Both Raf and ERK (extracellular signal-regulated kinase) molecules belong to the MAPK pathway [22]. It was reported that ERK was required for virus replication [23]. Our previous study indicated that AVL enhanced virus replication by activating ERK in HCT116 cells [15]. Unexpectedly, the present study showed that the level of phosphorylated ERK (p-ERK) in oncoVV-AVL infected cells was lower than that in oncoVV infected cells (Figure 5a). Thus, the expression of c-Raf was further detected in Hela S3 cells. As shown in Figure 5a, the expression of phosphorylated c-Raf (p-c-Raf) in oncoVV-AVL infected cells was significantly higher than that in oncoVV infected cells.

To further explore the underlying mechanism, a Raf inhibitor (Sorafenib) or an ERK inhibitor (U0126) was combined with the oncolytic vaccinia virus to treat Hela S3 cells. Subsequently, the virus replication was measured. The results indicated that sorafenib supplementation significantly improved the replication ability of oncoVV-AVL but not oncoVV (Figure 5b). We also found that the addition of U0126 could only slightly enhance the virus replication (Figure 5c). These indicated that AVL harboring interfered with the Raf/ERK pathway in Hela S3 cells, which was different from the general tumor cells we studied before. Furthermore, according to the exhibition of AVL, Raf and Sorafenib, we deduce that there may be a certain feedback regulation signal between the AVL and Raf/ERK pathway.

### 2.6. OncoVV-AVL Inhibited Tumor Growth in Mice

To assess the antitumor effect of oncoVV-AVL on Hela S3 cells in vivo, xenograft tumor models were established in Balb/c nude mice by subcutaneous injection of Hela S3 cells. Then oncoVV, or oncoVV-AVL, or the same volume of saline was injected intratumorally when tumor volume reached 100~200 mm^3^. As shown in Figure 6a,b, oncoVV-AVL exhibited better antitumor effects than oncoVV.

To demonstrate the distribution of viruses in vivo, immunohistochemistry was performed by using a primary antibody against oncolytic vaccinia viruses (A27L). The result showed that the tumor injected with oncoVV-AVL expressed a higher level of viral A27L protein than NaCl and oncoVV controls (Figure 6c). These indicated that oncoVV-AVL had a higher replication rate in xenograft tumors.

In addition, to further confirm the results of the in vitro experiment, OASL protein was examined in xenograft tumors. Similar to A27L, the tumors infected with oncoVV-AVL expressed a higher level of OASL than NaCl and oncoVV controls. Particularly, the expression of OASL was detected at the cell membrane (Figure 6d). Therefore, we conclude that AVL harboring may enhance the virus replication by stimulating the expression of OASL.

## 3. Discussion

Traditionally, an in vitro cell-based assay is carried out using two-dimensional cell culture. However, most tumor cells exist in a three-dimensional (3D) microenvironment. The phenotype and function of cells are strongly dependent on the interactions with neighboring cells when cultured in a 3D system [24]. Some characteristics of cells that are critical for physiological cell-based assays could be recovered in a 3D culture system. Therefore, cells are induced into free-floating spheroids to maintain the characteristics [25,26]. Initially, the free-floating sphere was first described in a brain tumor [27]. In the ensuing years, tumorospheres were developed from a wide range of tumors [28,29,30] under the assumption that “sphere assays” enable measuring self-renewal capacity [26]. Here, we addressed that AVL enhanced the killing efficiency of oncolytic vaccinia virus by enhancing virus replication in Hela S3 cells, a cell line capable of sphere formation. Therefore, we provide a novel strategy for cancer therapy, especially for the tumor cells with stemness.

The MAPK signaling pathway regulates a variety of cellular functions that are important for tumorigenesis, which consists of at least four main components, including Ras, Raf, MEK and ERK [31]. Ras is a small GTPase that is held at the inner surface of the plasma membrane, being functionally similar to the Gα subunit of the G protein. Raf is the best characterized as a Ras effector. Following the activation of Ras, Raf is recruited to the cell membrane. Raf activation stimulates a signaling cascade by phosphorylation of downstream proteins, such as ERK1 and ERK2. Subsequently, ERKs phosphorylate can activate a variety of nuclear transcription factors and kinases. In our research, AVL harboring enhanced the level of p-c-Raf but decreased the expression of c-ERK. Furthermore, the supplementation of Raf inhibitors obviously enhanced the virus replication. These suggested that AVL interfered with the Raf/ERK pathway through an unknown relationship, which deserves further exploration.

The activation of the JAK/STAT pathway stimulates the formation of interferon stimulated gene factor 3 (ISGF3). Subsequently, ISGF3 enters the nucleus and binds to IFN stimulated response element (ISRE), which then initiates the expression of the IFN stimulated genes (ISGs). The ISGs family includes many members, such as OASL. This study showed that AVL harboring enhanced the expression of OASL and thus elevated the replication of oncolytic vaccinia virus. Ghosh had reported that OASL inhibited IFN induction both in vivo and in vitro during DNA virus infection, which was opposite to its IFN-promoting antiviral activity against RNA virus infection [21]. Our results are consistent with Ghosh’s study. Therefore, we summarize that AVL harboring enhanced the antitumor effect of oncolytic vaccinia virus by promoting virus replication, which was related to higher expression of OASL. OASL may work as a negative-feedback regulator to inhibit IFN induction; in this study, this instead elevated the virus replication.

## 4. Materials and Methods

### 4.1. Cell Lines and Cell Culture

The human embryonic kidney cell line HEK293A, Human cervical cancer cell line Hela and Hela S3 were preserved in the laboratory. Cells were cultured in DMEM medium (Gibco, Thermo Fisher Scientific, Waltham, MA, USA) supplemented with 10% fetal bovine serum (Hyclone Laboratories, Dunedin, Otago, New Zealand) and 1% Penicillin-Streptomycin. The cells were grown at 37 °C in a humidified atmosphere containing 5% CO_2_.

For cell spheroidization, Hela and Hela S3 cells were cultured in serum free Advanced DMEM/F12 medium supplemented with 20 ng/mL EGF, 40 ng/mL b-FGF (PeroTech, Rocky Hill, NJ, USA), 2% B-27 and 1% N-2. The above reagents were purchased from Gibco Company (Thermo Fisher Scientific, Waltham, MA, USA) if not specified. Cells were seeded into 6-well ultra-low attachment plates (NEST) at the dose of 2 × 10^3^ cells/well in 1 mL sphere media with replicates of 3 per dose. When the spheres were observed under microscope, oncoVV or oncoVV-AVL (developed from vaccinia virus Western Reserve strain) at the concentration of 5 MOI was added to the wells, respectively. After 5 days of viral infection, the spheres with diameter ≥50 μm were counted.

### 4.2. Cell Proliferation and Cell Apoptosis Detection

Hela S3 cells (5 × 10^3^ cells/well) were transferred in five replicates to 96-well plates in 100 μL medium. All the cells were incubated at 37 °C in 5% CO_2_ for 12 h to allow the cells to attach to the bottom of the well. Serial dilution of oncoVV-AVL or oncoVV (1, 2, 5 and 10 MOI) was added and the control group was added with an equal dilution of PBS. At the culture time of 24, 48, 72 and 96 h, cells viabilities were detected by MTT assay (Beyotime Institute of Biotechnology, Shanghai, China). 20ul MTT (5 mg/mL) was added to wells and incubated at 37 °C for 4 h, then the medium was carefully removed. Then 150 μL DMSO was added. Finally, the absorbance was determined using an Enzyme mark instrument at the wavelength of 490 nm.

Flow cytometric analysis was used to detect apoptotic cells. Cells (4 × 10^5^ cells/well) were seeded into 6-well plates overnight to attach the bottom, then PBS, 5MOI oncoVV or oncoVV-AVL was added. After treatment with virus or PBS, cells were collected and stained with Annexin V-FITC and propidium iodide (PI) (BD Biosciences, San Jose, CA, USA) following the manufacturer’s instruction. Subsequently, the cells were analyzed by flow cytometry (AccuriC6, BD Biosciences, San Jose, CA, USA).

### 4.3. Virus Replication Assay

To measure the replication of the virus in Hela S3, cells (5 × 10^4^/well) were plated in 24-well plates and incubated overnight to attach to the bottom of the wells. Subsequently, 5 MOI virus or equal dilution of PBS was added. At the treatment time of 0 h, 12 h, 24 h, 36 h and 48 h, the cells and culture medium were collected and stored at −80 °C for the following experiments. The viral titers were determined by TCID50 assay in HEK293A cells.

### 4.4. Western Blot Analysis

Hela S3 cells were harvested and resuspended in lysis buffer (Beyotime Institute of Biotechnology, Shanghai, China). Samples of cytosolic proteins were separated by 10% SDS-PAGE and then transferred to a PVDF membrane (Millipore, Bedford, MA, USA). Blots were blocked with 5% non-fat milk at room temperature and then incubated with primary antibodies (1:1000 dilution) at 4 °C for overnight. After washing, the blots were incubated with secondary antibodies (1:5000 dilution) for 1 h at room temperature. The blots were detected using Clinx 6000EXP chemiluminescence image system (Clinx, Shanghai, China). The following antibodies were used: ERK, c-ERK, c-Raf, p-c-Raf and OASL, which were purchased from Cell Signaling Technology.

### 4.5. Xenograft Tumor Model in Immunodeficient Mice

Female Balb/c nude mice, at the age of 5–6 weeks, were purchased from Slack Animal Laboratory (Shanghai, China) and were housed (4 mice/cage) under standardized temperature (18–23 °C) with 50% humidity. The mice experiment was performed with the approval of the Experimental Animal Committee of Zhejiang Sci-Tech University. Hela S3 cells were injected into the back of mice subcutaneously at the dose of 5 × 10^6^ cells/mouse. When the tumors had grown to 100–200 mm^3^, the mice were randomly divided into 3 groups (6–8 mice/group). Subsequently, 1 × 10^7^ plaque-forming units (PFU) oncolytic vaccine virus or saline control was injected into the mice intratumorally. The volume of the tumors was measured every 4 days. The tumor volume was calculated according to the following formula: V (mm^3^) = length (mm) × width (mm)^2^ × 0.5. 5 or 6 weeks later, mice were sacrificed, and tumors were harvested and then fixed with 4% paraformaldehyde.

### 4.6. Immunohistochemistry

Paraffin-embedded tumor tissue slides were deparaffinized with xylene rehydrated in grades alcohols for further immunohistochemistry staining. For antigen retrieval, slides were boiled in citric acid buffer (pH6.0) for 15 min. Peroxidase was blocked by using 3% H_2_O_2_ for 10 min at room temperature. The slides were incubated overnight with the primary antibody against OASL or A27L (1:150) at 4 °C. Subsequently, peroxidase and hematoxylin were employed to visualize the staining. Negative control was used by omitting the primary antibody.

### 4.7. Statistical Analysis

The unpaired t-test was performed using GraphPad Instat software. All results were presented as means ± SEM, *p* < 0.05 was considered as statistically significant.

## 5. Conclusions

Firstly, the present study compared the ability of sphere formation between Hela and Hela S3 cells. It then demonstrated that a newly constructed oncoVV-AVL induced Hela S3 cells death significantly both in vitro and in vivo. The antitumor mechanisms of oncoVV-AVL may be related to the activation of the Ras/ERK pathway and the higher expression of OASL. These findings may provide insight into oncolytic viral therapies armed with AVL.

## Figures and Tables

**Figure 1 marinedrugs-19-00532-f001:**
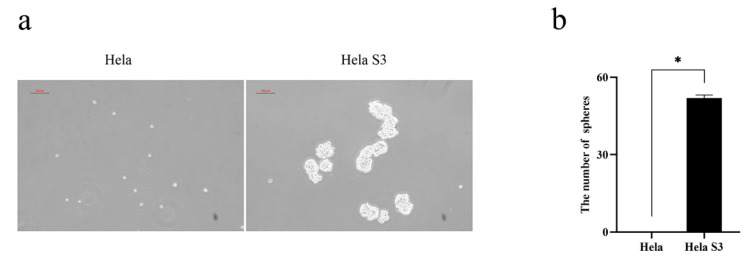
Tumorospheres formation assay of Hela and Hela S3 cells. (**a**) Hela S3 cells were more capable of spheroidization than Hela cells; (**b**) quantification of tumorospheres of Hela and Hela S3 cells. Data were expressed as mean ± SEM from three independent experiments (* *p* < 0.05).

**Figure 2 marinedrugs-19-00532-f002:**
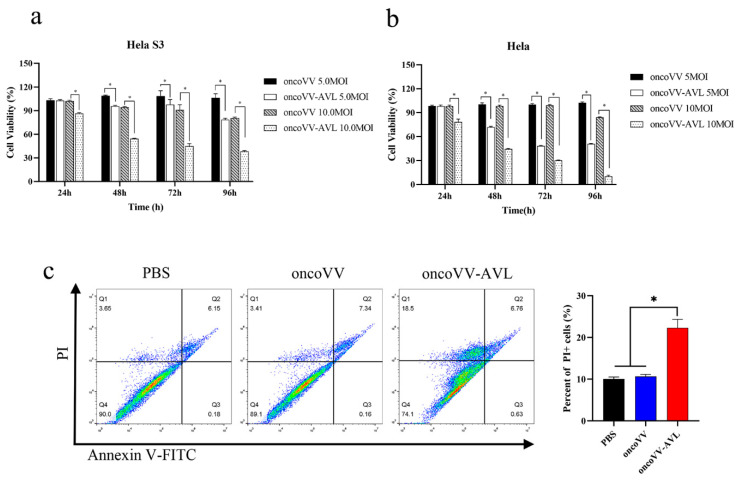
The cytotoxic effect of oncoVV and oncoVV-AVL on Hela S3 and Hela cells. (**a**) Cell viability was measured by MTT assay in Hela S3 cells; (**b**) cell viability was measured by MTT assay in Hela cells; (**c**) the percentage of apoptotic/dead cells was detected using flow cytometry. Hela S3 cells were treated with PBS, or oncoVV, or oncoVV-AVL for 48 h, respectively, then cells were stained by Annexin V-FITC/PI followed by detection under flow cytometry. Data were expressed as mean ± SEM from three independent experiments. (* *p* < 0.05).

**Figure 3 marinedrugs-19-00532-f003:**
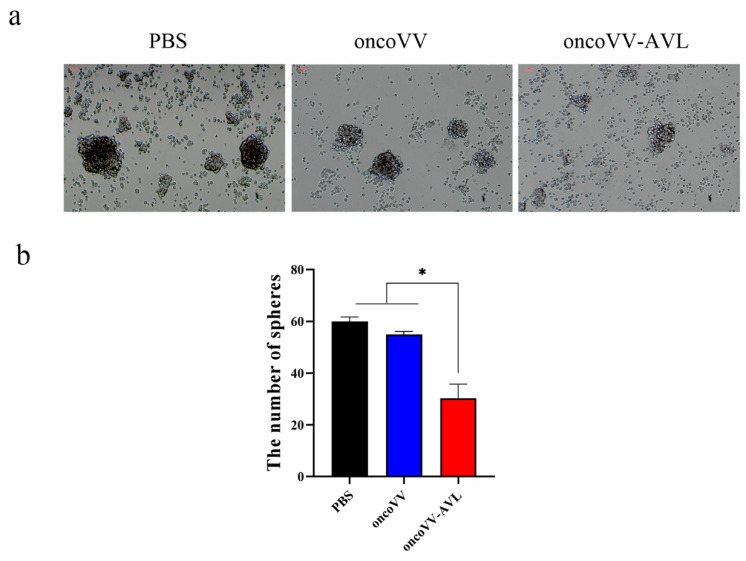
The cytotoxic effect of oncoVV-AVL on tumoropheres in vitro. (**a**) Morphology of the tumorospheres; (**b**) quantification of the tumorospheres. Data were presented as the mean ± SEM of three independent experiments. (* *p* < 0.05).

**Figure 4 marinedrugs-19-00532-f004:**
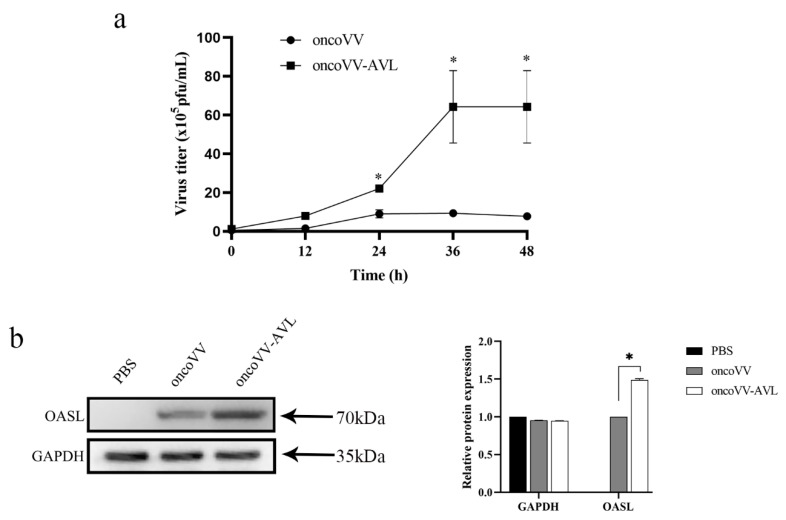
AVL enhanced virus replication by promoting the expression of OASL. (**a**) AVL improved the replication of oncolytic vaccinia virus in Hela S3 cells; oncoVV or oncoVV-AVL was used at the concentration of 5 MOI, virus titers were determined by TCID50 assay in HEK293A cells. (**b**) The expression of OASL protein was detected by Western blot, GAPDH served as the loading control. Data were presented as the mean ± SEM from at least three independent experiments. (* *p* < 0.05).

**Figure 5 marinedrugs-19-00532-f005:**
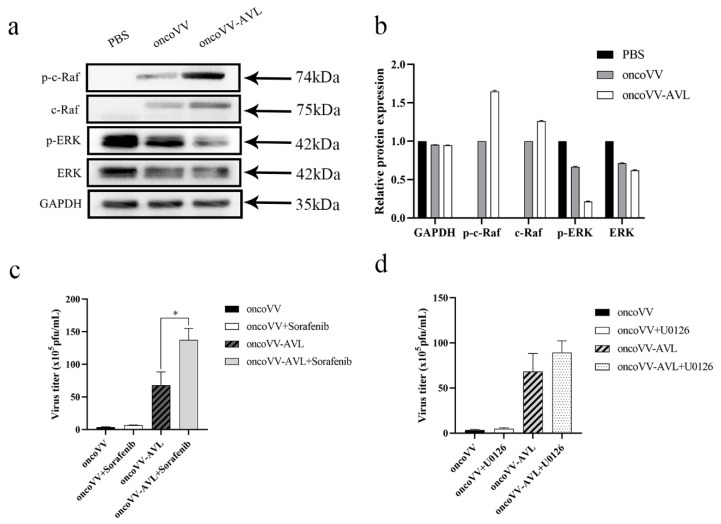
AVL stimulated the Raf /ERK signaling pathway in Hela S3 cells. (**a**) The expressing levels of c-Raf, p-c-Raf, ERK and p-ERK were detected by Western blot; (**b**) Quantification of Western blot; (**c**)the virus titers were determined in Hela S3 cells treated with Sorafenib; (**d**) the virus titers were determined in Hela S3 cells treated with U0126. Virus titers were determined by TCID50 assay in HEK293A cells. Data were presented as the mean ± SEM from at least three independent experiments. (* *p* < 0.05).

**Figure 6 marinedrugs-19-00532-f006:**
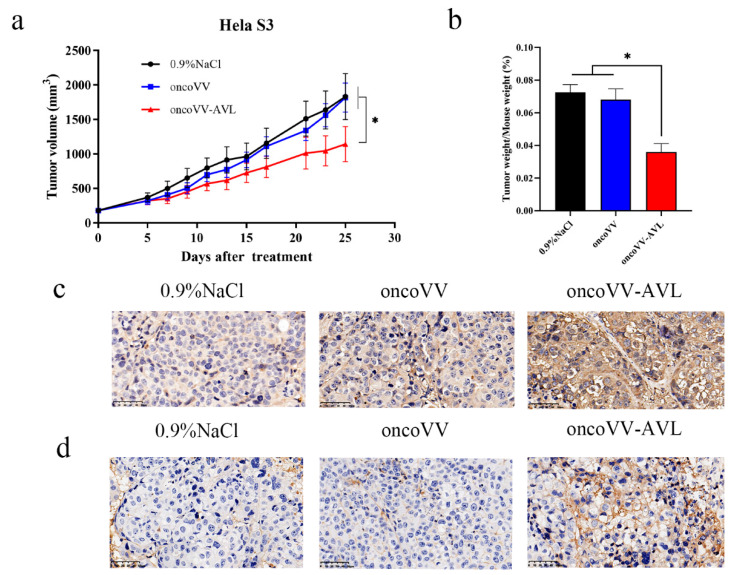
The antitumor effects of oncoVV-AVL in mice. (**a**) Hela S3 cells were inoculated into the Balb/c nude mice on the back, the mice were then injected with oncoVV, or oncoVV-AVL, or saline control intratumorally. The volume of tumor was measured every 4 days; (**b**) ratio of tumor weight to mouse body; immunohistochemical analysis showed that oncoVV-AVL enhanced the level of A27L (**c**) and OASL (**d**) in xenograft tumor of Hela S3 cells. Scale bars show 50 µm. Data were presented as the mean ± SEM from at least three independent experiments. (* *p* < 0.05).

## Data Availability

Data are contained within the article.

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
