# Peer review of "Oncolytic Vaccinia Virus Harboring Aphrocallistes vastus Lectin Inhibits the Growth of Cervical Cancer Cells Hela S3"

_marinedrugs, 2021, doi:10.3390/md19100532_

Round 1

Reviewer 1 Report

The authors in the manuscripts titled “ Oncolytic vaccinia virus harboring achronalities vastus lectin  inhibits the growth of cervical cancer cell Hela S3” have outlined their findings of onco VV-AVL on hela S3 cells and proposed that it could be a novel technique to utilize lection AVL. While the authors have provided basic evidence for the use of the virus they still need to further provide evidence to support the claim

  1. The current results are demonstrated in only one cell line. This could be a biased interpretation of the results. In order for the virus to be accepted as a true oncolytic virus the authors should present the key findings of their study in other aggressive cell lines as well.
  2. The authors need to perform neutralization assay with the said virus to demonstrate its effectiveness.
  3. In the in vivo experiment, could the authors demonstrate the presence of the virus in the tumor? Alternatively, can the virus be sequestered in the liver? Without the evidence that the virus actually is present actively in the tumor, current result doesn’t hold much merit.
  4. Does the virus replicate in vivo n nude mice? Alternatively, could the authors perform similar experiment in an immune competent model? For eg: Hamster?

Author Response

The authors in the manuscripts titled “ Oncolytic vaccinia virus harboring achronalities vastus lectin  inhibits the growth of cervical cancer cell Hela S3” have outlined their findings of onco VV-AVL on hela S3 cells and proposed that it could be a novel technique to utilize lection AVL. While the authors have provided basic evidence for the use of the virus they still need to further provide evidence to support the claim

  1. The current results are demonstrated in only one cell line. This could be a biased interpretation of the results. In order for the virus to be accepted as a true oncolytic virus the authors should present the key findings of their study in other aggressive cell lines as well.

Response: We thank the reviewer for very careful check and constructive comments. We have added Hela cells to verify the antiproliferative effect on cervical cancer as shown in Fig2b.

  1. The authors need to perform neutralization assay with the said virus to demonstrate its effectiveness.

Response: We agree with the reviewer that neutralization assay will help to demonstrate the effectiveness of the viruses. However, currently no commercially available antibodies are suitable for neutralization. In our paper, we performed MTT assay, flow cytometry analysis, as well as animal experiments to exam the effectiveness of oncoVV-AVL. All our results showed that oncoVV-AVL elicited significant suppressive effect on cervical cancer cells.

  1. In the in vivo experiment, could the authors demonstrate the presence of the virus in the tumor? Alternatively, can the virus be sequestered in the liver? Without the evidence that the virus actually is present actively in the tumor, current result doesn’t hold much merit.

Response: As shown in our in vivo experiment Fig6c, the present of vaccinia viruses was investigated through Immunohistochemical analysis for viral A27L proteins.

  1. Does the virus replicate in vivo n nude mice? Alternatively, could the authors perform similar experiment in an immune competent model? For eg: Hamster?

Response: We showed the spreading of the virus in xenograft tumors as shown in our animal studies, suggesting that the virus replicated in the in vivo model.

Reviewer 2 Report

Manuscript “Oncolytic vaccinia virus harboring aphrocallistes vastus lectin inhibits the growth of cervical cancer cell Hela S3” represents interesting data on the effect of insertion of aphrocallistes vastus lectin gene into genome of vaccinia virus. Although the overall impression of the manuscript is positive, the text needs improvement.

              First, the text shows insufficient knowledge of virology, which is reflected, first of all, in the terminology used. Thus, “oncolytic vaccinia virus” is not systematic name, it is necessary to present name of the strain and source of a virus. It is generally accepted to use a combination of the strain name and "vaccinia virus" for an abbreviation, for example: LIVP-VacV or LIVP-VV. Using of “oncoVV-AVL” and “oncoVV” terms is incorrect, and I strongly recommend to improve this. 

               I am puzzled, why the article lacks information on the preparation of the recombinant, its virological and molecular characteristics, and this information is absolutely necessary. I do not think that its absence can be justified by the fact that the "Marine drugs" journal is not virological. Include the information, please. You may use published articles doi: 10.3390/v8010020 and doi: 10.18632/oncotarget.2579, as an example how to describe a virus with oncolytic properties.

               The next comment concerns the term “steamness”. In section 2.1 we see: “Previous studies suggested that sphere-forming activity represented the stemness of the cells to a certain extent [16]”. This is interesting information and I moved to corresponding publication:

  1. Stein, M.N.et al. J.M. First-in-Human Clinical Trial of Oral ONC201 336 in Patients with Refractory Solid Tumors. Clinical Cancer Research 2017, 23(15), 4163-4169.

I was really surprised: it was human study, absolutely no information about spheres and steamness!

“The design was an open-label, dose escalation phase I trial of monoagent ONC201 in patients with advanced, refractory solid tumors who had exhausted or refused standard treatment options for their respective indications”.

I don’t want to think that the authors "invented" this reference, perhaps this is a confusion with citations. I cannot verify the correctness of all the cited works, the authors must carry out this work.

Returning to the term "steamness": the cited fact of the dependence of steamness and the formation of spheres is interesting, and I want to see serious confirmation.

The authors use the term "antitumor" throughout the text. However, this term can only be applied to the results of animal experiments, when a tumor is present. MTT analysis data cannot in any way correlate with the antitumor effect. (section 2.2). Please check the correctness of this term and other used, and apply terms that strictly correspond to the meaning of the study.  

Information on experimental design of spheres incubation with viruses is missed in Materials and methods, it must be included. Pay attention at time-point of spheres formation, and parameters of virus treatment, justify seeding dose for spheres formation.

The text of section 2.3 is not clear about the results obtained. No description of spheres before treatment with virus; no size characteristics of spheres.  The conclusive sentence of this section is not supported by the results: “This demonstrated that oncoVV-AVL inhibited the growth of tumorospheres in Hela S3 cell, suggested that oncoVV-AVL may be used as an agent to attack the tumor stem cells in the future”. Authors should critically analyze their results in light of published data concerning tumorospheres formation.

Another incorrect term is "to promote" in relation to the virus (line 157, 163 etc.), replace it with an adequate meaning.

The authors continue to amaze with a careless attitude to the terminology and meaning of words. What does it mean: Particularly, the expression of OASL was very strong in the region of virus infection, where exhibited many viral infection plaques (line 162). What plaques can be observed in tumor tissue?

line 214: HEK293A – where this line was used?

Section 4.2. Cell proliferation and cell apoptosis detection: include parameters of cells and viruses incubation.

Author Response

Manuscript “Oncolytic vaccinia virus harboring aphrocallistes vastus lectin inhibits the growth of cervical cancer cell Hela S3” represents interesting data on the effect of insertion of aphrocallistes vastus lectin gene into genome of vaccinia virus. Although the overall impression of the manuscript is positive, the text needs improvement.

              First, the text shows insufficient knowledge of virology, which is reflected, first of all, in the terminology used. Thus, “oncolytic vaccinia virus” is not systematic name, it is necessary to present name of the strain and source of a virus. It is generally accepted to use a combination of the strain name and "vaccinia virus" for an abbreviation, for example: LIVP-VacV or LIVP-VV. Using of “oncoVV-AVL” and “oncoVV” terms is incorrect, and I strongly recommend to improve this. 

Response: We agree with the reviewer that “oncolytic vaccinia virus” is not a systematic name. However, most known oncolytic vaccinia viruses are not named using strain names, such as famous JX-594 (Wyeth strain) and vvDD (Western reserve strain) which have entered clinical trials. OncoVV-AVL has been used in our previous publication, to avoid confusing we suggest to keep using this name.

               I am puzzled, why the article lacks information on the preparation of the recombinant, its virological and molecular characteristics, and this information is absolutely necessary. I do not think that its absence can be justified by the fact that the "Marine drugs" journal is not virological. Include the information, please. You may use published articles doi: 10.3390/v8010020 and doi: 10.18632/oncotarget.2579, as an example how to describe a virus with oncolytic properties.

Response: We agree with the reviewer that the recombination process and production of our virus should be presented, although it has been described in our previous AVL paper. In introduction section, L36-38, we added: “In our previous study, a gene encoding AVL was inserted into an oncolytic vaccinia virus (oncoVV) vector, which is deficient of TK gene for cancer specific replication, forming a recombinant virus oncoVV-AVL.”

               The next comment concerns the term “steamness”. In section 2.1 we see: “Previous studies suggested that sphere-forming activity represented the stemness of the cells to a certain extent [16]”. This is interesting information and I moved to corresponding publication:

  1. Stein, M.N.et al. J.M. First-in-Human Clinical Trial of Oral ONC201 336 in Patients with Refractory Solid Tumors. Clinical Cancer Research 2017, 23(15), 4163-4169.

I was really surprised: it was human study, absolutely no information about spheres and steamness!

“The design was an open-label, dose escalation phase I trial of monoagent ONC201 in patients with advanced, refractory solid tumors who had exhausted or refused standard treatment options for their respective indications”.

I don’t want to think that the authors "invented" this reference, perhaps this is a confusion with citations. I cannot verify the correctness of all the cited works, the authors must carry out this work.

Returning to the term "steamness": the cited fact of the dependence of steamness and the formation of spheres is interesting, and I want to see serious confirmation.

Response: We thank the reviewer for very careful check. We have carefully checked the citations in this paper and corrected the wrong citation on spheres. (L53)

The authors use the term "antitumor" throughout the text. However, this term can only be applied to the results of animal experiments, when a tumor is present. MTT analysis data cannot in any way correlate with the antitumor effect. (section 2.2). Please check the correctness of this term and other used, and apply terms that strictly correspond to the meaning of the study.  

Response: We have corrected according to the reviewer’s comment.

Information on experimental design of spheres incubation with viruses is missed in Materials and methods, it must be included. Pay attention at time-point of spheres formation, and parameters of virus treatment, justify seeding dose for spheres formation.

The text of section 2.3 is not clear about the results obtained. No description of spheres before treatment with virus; no size characteristics of spheres.  The conclusive sentence of this section is not supported by the results: “This demonstrated that oncoVV-AVL inhibited the growth of tumorospheres in Hela S3 cell, suggested that oncoVV-AVL may be used as an agent to attack the tumor stem cells in the future”. Authors should critically analyze their results in light of published data concerning tumorospheres formation.

Response: L227-229, we added: When the spheres were observed under microscope, oncoVV or oncoVV-AVL at the concentration of 5 MOI was added to the wells respectively. After 5 days of viral infection, the spheres with diameter ≥ 50 μm were counted.

L90-91, we revised: “This demonstrated that oncoVV-AVL inhibited the growth of tumorospheres of Hela S3 cells.”

Another incorrect term is "to promote" in relation to the virus (line 157, 163 etc.), replace it with an adequate meaning.

Response: We have revised according to the reviewer’s comment.

The authors continue to amaze with a careless attitude to the terminology and meaning of words. What does it mean: Particularly, the expression of OASL was very strong in the region of virus infection, where exhibited many viral infection plaques (line 162). What plaques can be observed in tumor tissue?

Response: We have replaced “plaques” with “vesicles”.

line 214: HEK293A – where this line was used?

Response: We corrected virus replication assay in materials and methods section, L251, in which “293A” was replaced with “HEK293A”.

Section 4.2. Cell proliferation and cell apoptosis detection: include parameters of cells and viruses incubation.

Response: We revised according to the reviewer’s comments in L236-242.

Reviewer 3 Report

The presented manuscript is a logical continuation of the work of the same authors on the study of the antitumor properties of recombinant vaccinia virus expressing AVL. The previous studies demonstrated that gene encoding AVL enhanced the inhibitory effect of oncolytic vaccinia virus in a variety of cancer cells. However, among the studied cancer cells, there are no Hela cells, the derivative of which Hela S3 they use in the present study. Nevertheless, it is of great interest to compare the inhibitory properties of oncoVV-AVL both in relation to the initial heterogeneous population of Hela cells and in relation to Hela S3 cells, which the authors use as a model of tumor stem cells.

I have no major amendments, but there are a number of editorial comments:

  • In Figures 4 and 5, the virus titer is indicated in pfu/mL, while the figure legends and the M&Ms indicate that the virus titers were determined by TCID50 assay. Please, correct TCID50 assay to Plaque assay.
  • It is necessary to correct the legend to Figure 6c - there are contradictions.
  • It is necessary to carefully check the text of the manuscript, which contains inaccuracies, for example, the use of the term “cell” instead of “cells”, including title of the paper.

Author Response

The presented manuscript is a logical continuation of the work of the same authors on the study of the antitumor properties of recombinant vaccinia virus expressing AVL. The previous studies demonstrated that gene encoding AVL enhanced the inhibitory effect of oncolytic vaccinia virus in a variety of cancer cells. However, among the studied cancer cells, there are no Hela cells, the derivative of which Hela S3 they use in the present study. Nevertheless, it is of great interest to compare the inhibitory properties of oncoVV-AVL both in relation to the initial heterogeneous population of Hela cells and in relation to Hela S3 cells, which the authors use as a model of tumor stem cells.

I have no major amendments, but there are a number of editorial comments:

  • In Figures 4 and 5, the virus titer is indicated in pfu/mL, while the figure legends and the M&Ms indicate that the virus titers were determined by TCID50 assay. Please, correct TCID50 assay to Plaque assay.

Response: We thank the reviewer for very careful check. Although TCID50 assay is different from plaque assay, it also determines viral titers through counting plaque formation in each well and get pfu/ml unit.

  • It is necessary to correct the legend to Figure 6c - there are contradictions.

Response: We revised: “Immunohistochemical analysis showed that oncoVV-AVL enhanced the level of A27L (c)”

  • It is necessary to carefully check the text of the manuscript, which contains inaccuracies, for example, the use of the term “cell” instead of “cells”, including title of the paper.

Response: We have revised according to the reviewer’s comment.

Round 2

Reviewer 2 Report

The authors have made changes in accordance with the recommendations. However, it is not clear to me why it is impossible to indicate the virus strain in the "Materials and Methods" section. It cannot be that the authors did not know it if they themselves prepared the recombinant. I insist on specifying the strain.

Then, figure 6 d does not show vesicles! This term is incorrect. Please, ask somebody to provide histologically correct legend. 

I also recommend that you read the text carefully once again, the meaning of the words should correspond to the facts. Thus, "highest mortality rates" (line 68) does not apply to cells.

Author Response

The authors have made changes in accordance with the recommendations. However, it is not clear to me why it is impossible to indicate the virus strain in the "Materials and Methods" section. It cannot be that the authors did not know it if they themselves prepared the recombinant. I insist on specifying the strain.

Response: In the materials and methods section, L228, we added: (developed from vaccinia virus Western Reserve strain).

Then, figure 6 d does not show vesicles! This term is incorrect. Please, ask somebody to provide histologically correct legend. 

Response: L165-166, we revised: the expression of OASL was detected at cell membrane.

I also recommend that you read the text carefully once again, the meaning of the words should correspond to the facts. Thus, "highest mortality rates" (line 68) does not apply to cells.

Response: We thank the reviewer for very careful check. L70-71, we revised: compared with oncoVV and PBS controls, oncoVV-AVL induced a significantly higher level of cytotoxicity.